# Phylogenomics and antimicrobial resistance of the leprosy bacillus *Mycobacterium leprae*

Andrej Benjak et al.[#] (iD)

Leprosy is a chronic human disease caused by the yet-uncultured pathogen *Mycobacterium leprae*. Although readily curable with multidrug therapy (MDT), over 200,000 new cases are still reported annually. Here, we obtain *M. leprae* genome sequences from DNA extracted directly from patients' skin biopsies using a customized protocol. Comparative and phylogenetic analysis of 154 genomes from 25 countries provides insight into evolution and antimicrobial resistance, uncovering lineages and phylogeographic trends, with the most ancestral strains linked to the Far East. In addition to known MDT-resistance mutations, we detect other mutations associated with antibiotic resistance, and retrace a potential stepwise emergence of extensive drug resistance in the pre-MDT era. Some of the previously undescribed mutations occur in genes that are apparently subject to positive selection, and two of these (*ribD*, *fadD9*) are restricted to drug-resistant strains. Finally, nonsense mutations in the *nth* excision repair gene are associated with greater sequence diversity and drug resistance.

. Correspondence and requests for materials should be addressed to S.T.C. (email: stewart.cole@epfl.ch). [#]A full list of authors and their affliations appears at the end of the paper.

**M**ycobacterium leprae is the main causative agent of leprosy, a disease that affects the skin, nerves, and mucosa of the upper respiratory tract in humans[1]. A second, distantly related leprosy bacillus, Mycobacterium lepromatosis, was recently discovered in humans and red squirrels (Sciurus vulgaris)[2]. Leprosy is curable with multidrug therapy (MDT), but remains a public health problem in South America, Africa, South and Southeast Asia, and Micronesia, where over 200,000 new leprosy cases are reported each year[3]. MDT, comprising rifampicin, dapsone, and clofazimine, has been used intensively since the 1980s and a few second-line drugs, ofloxacin, minocycline, and clarithromycin, are sometimes employed as therapeutic agents[4]. The emergence of drug-resistant (DR) and multidrug-resistant (MDR) M. leprae is increasingly reported[5–12]. For dapsone, rifampicin and ofloxacin, the resistance mechanism has been attributed to missense mutations in the drug resistance determining regions (DRDR) of the folP1, rpoB, and gyrA genes, respectively.

M. leprae is an obligate intracellular pathogen that has never been cultured axenically but can infect wild or experimental animals. The nine-banded armadillo (Dasypus novemcinctus) or the mouse footpad (MFP) can be used to produce bacilli, but both methods are cumbersome and time-consuming[13]. The genome of M. leprae is the smallest among mycobacteria (3.3 Mb) with 1614 genes encoding proteins and a remarkable 1300 pseudogenes[14]. Such reductive evolution is a hallmark of bacteria that have changed their lifestyle from free-living to strictly host-associated[15]. Due to its 14-day generation time and the absence of horizontal gene transfer, the genome of M. leprae is highly conserved, with <300 single-nucleotide polymorphisms (SNPs) observed between distantly related strains, and only a few SNPs between close relatives[5,16–18]. Four SNP types (branches 1–4) and 16 SNP subtypes (A–P) were defined by surveying 78 informative SNPs and six single-base insertion/deletions (InDels)[16,19]. Genotyping a large panel of M. leprae strains revealed strong geographical associations and suggested possible routes of dissemination of leprosy[16] whereas, a recent phylogenetic analysis of 16 whole-genome sequences of modern and ancient M. leprae strains, implicated the 3K subtype (branch 0) as the most ancestral[17].

Leprosy seems to have appeared during the Iron Age (1200–600 BC) and the date of the most recent common ancestor of M. leprae was estimated to be from 2543 BC to 36 AD, based on whole-genome sequence analysis[17]. Similarly, the earliest accepted written record of leprosy is from 600 BC[20], and the earliest osteological evidence dates from around 300 BC[21–25]. The oldest genomic evidence of leprosy is for samples from 80 to 240 AD in Central Asia[26].

In this study, we develop and apply methods to isolate and purify M. leprae DNA that enable whole genome sequences to be obtained directly from human biopsy material, thus removing the necessity for passage through animals. This approach was successfully used to generate 120 new M. leprae genome sequences from drug-susceptible and DR strains from around the world, thereby enabling detailed phylogenetic and phylogeographic comparisons to be performed, new mutations associated with antimicrobial resistance to be detected, and the likely origin of leprosy to be proposed.

## Results
### Isolating M. leprae DNA from human skin biopsies.
Genome sequencing has become routine practice in microbiology[27], especially for micro-organisms that can be readily isolated, which is not the case of the leprosy bacillus. For decades, the sole source of M. leprae DNA suitable for genomics was from bacteria

isolated 12 months after infection of armadillos or mice. Recently, we have developed and optimized methods that enable M. leprae DNA to be extracted directly from fresh or formalin-fixed skin biopsies from leprosy patients[28]. These methods include enrichment of M. leprae DNA by array capture[17] but this is less practical for large population-based investigations.

The DNA extraction method used in this study was applied directly to punch biopsies from clinically well-characterized patients of known bacillary index (BI) and exploits the fact that M. leprae resides intracellularly. Host cells are first disrupted and their DNA degraded, leaving the bacilli intact. The bacilli are then lysed and their DNA extracted and used for library preparation. This approach was applied to 106 biopsies whose BI ranged from 0 (no bacilli visible) to 6 (>1000 bacilli per microscopic field) thereby enabling a relationship between BI and sequencing efficiency to be established (Fig. 1). As expected, there was a direct correlation between genome coverage and the BI but, surprisingly, successful coverage could even be achieved with some specimens whose BI was as low as 1+.

**Genome analysis of patient and animal cohort.** We analyzed a total of 154 M. leprae genomes from 25 countries (Fig. 2, Supplementary Data 1), of which 120 were newly sequenced and 34 were previously published (Supplementary Data 1). The cohort comprised 147 human samples, 6 from red squirrels and 1 from an armadillo that were all naturally infected. Genome sequences were obtained directly from 109 human samples, 30 from bacilli passaged in mice, and 8 from armadillos. Thirty of these strains were from patients who had relapsed or not responded to MDT the remainder (124) were from supposedly drug-susceptible strains (87 were from confirmed primary cases, while disease history was unknown for the others).

A total of 3053 SNPs and 219 InDels (excluding tandem repeats) was found (Supplementary Data 2). The average SNP difference among the 154 genomes was 114. We found a total of 988 non-synonymous alleles (0.62 per protein-coding gene, or 0.61 per kb of protein-coding genes) and 530 synonymous SNPs (0.33 per protein-coding gene or 0.33 per kb of protein-coding genes), and 1763 mutations in intergenic regions and pseudogenes (1.07 mutations per kb of intergenic regions and pseudogenes). The SNP density for each gene is given in Supplementary Data 2. Of the 219 InDels, 58 (27%) were in protein-coding genes.

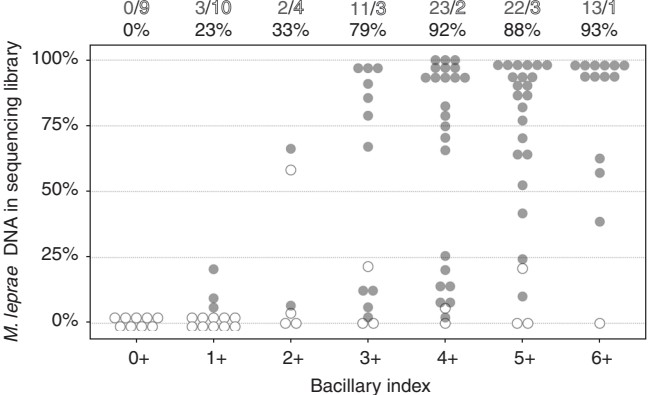

**Fig. 1** Correlation between bacillary index and successful sequencing. The content of M. leprae DNA in sequencing libraries derived from human skin biopsies was determined and found to be proportional to the bacillary index (not available for all samples). Empty circles are samples that were not included in the study due to insufficient genome coverage. Sample count and sequencing success rates are given at the top of each category

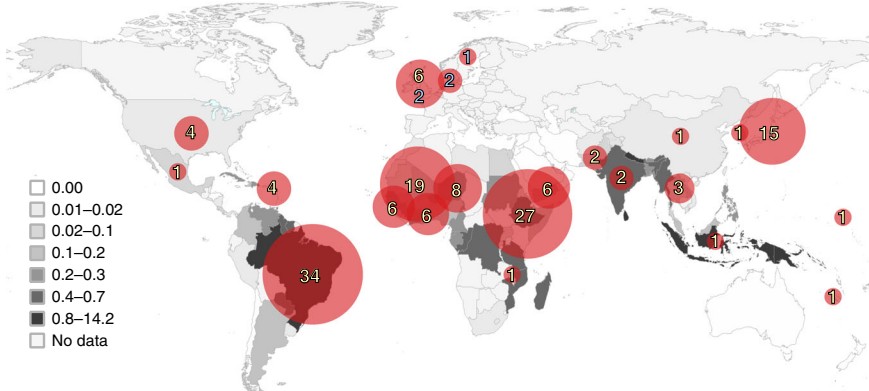

**Fig. 2** Geographic distribution of the *M. leprae* samples used in this study. World map shows the number of registered cases of leprosy per 10,000 population (prevalence rates) in 2015 as reported by the World Health Organization (http://apps.who.int/neglected_diseases/ntddata/leprosy/leprosy.html). Blue numbers indicate ancient *M. leprae* strains

**Phylogeny of *M. leprae*.** Phylogenetic analysis using both maximum parsimony (MP) and Bayesian inference resulted in consistent tree topologies and revealed distinct lineages and sublineages of *M. leprae* (Fig. 3). Strains belonging to the same SNP subtypes[16] clustered within single branches, with the exception of SNP subtype 3K, which is represented by a newly discovered ancestral lineage, termed here 3K-1, and the ancestral lineage referred to earlier as branch 0[17] and termed here 3K-0 (Fig. 3a). All strains from the two most ancestral lineages, 3K-0 and 3K-1, originated from Japan (8), China (1), Korea (1), the Marshall Islands (1), and New Caledonia (1), in agreement with earlier genotyping studies of hundreds of *M. leprae* strains which confirmed the predominance of the 3K genotype in East Asia, notably in Japan, China, and Korea[16,29–31].

*M. leprae* in East Africa showed higher diversity with subtypes 2E, 2F, and 2H representing distinct lineages (Fig. 3a). The geographic distribution of those lineages corroborates the findings of earlier studies reporting the presence of SNP type 2 in Medieval Europe, the Middle East and East Africa[16]. Two Ethiopian isolates, belonging to the 2F subtype, clustered closely with medieval European strains dating from the 11th to 12th century (Fig. 3), which supports the hypothesis that the ancient Greek and Roman routes[32] connecting Europe, the Middle East, East Africa, and South Asia[16,33–35] contributed to the dissemination of SNP type 2 *M. leprae*.

West Africa, on the other hand, harbors exclusively SNP type 4, suggesting that overland migration between East and West Africa was limited. SNP subtypes 4N, 4O, and 4P, albeit sharing the same ancestor, do not form a monophyletic clade as previously hypothesized[16]. Rather, the 4O and 4P subtypes cluster together in a branch distinct from 4N (Fig. 3).

Brazil, as expected, contains a great diversity of several *M. leprae* lineages, with the SNP type 4 and SNP subtype 3I being the most prevalent[36]. The 3I genotype was common in medieval Europe[17,21,37,38], and is still present in red squirrels in the United Kingdom[2]. The modern Brazilian strain Br2016-45 branched between two medieval strains from Europe (Fig. 3), making it the most ancestral contemporary 3I strain in the Americas to date. The broad diversity of 3I genotypes from Brazil probably derives from multiple introductions from Europe. On the other hand, the strains circulating in the Southern USA and associated with zoonosis from the nine-banded armadillo, I-30, NHDP-55 and NHDP-63[18], originated much more recently (Fig. 3b), in agreement with the rapid expansion and spread of the armadillo population since its introduction to this region about 150 years ago[39].

Good representation of most *M. leprae* lineages enabled identification of lineage-specific markers. A set of 235 SNPs and 25 InDels were specific to single lineages or groups of related lineages (Supplementary Data 2), of which 73 non-synonymous SNPs and 5 InDels were within protein-coding genes. These new lineage-specific markers can be used for future genotyping schemes.

**Dating analysis.** Dating analysis of *M. leprae* (Fig. 3b and Supplementary Figure 1) was done using BEAST v2.4.4[40] and the results were very similar to those obtained from a ten-fold smaller number of contemporary isolates[17]. The most recent common ancestor (TMRCA) of all *M. leprae* strains was estimated to be 3699 years old (95% Highest Posterior Density (HPD) 2731–4838 ya) and the substitution rate was $7.8 \times 10^{-9}$ per site per year. Overlapping results were obtained when using different models (Supplementary Note and Supplementary Table 1), indicating that the dataset was robust and sufficiently informative.

A striking observation is the relative youth of the SNP type 1 lineage and its association with South Asia (Fig. 3b). Earlier studies revealed a predominance of SNP subtype 1D in India and Nepal, followed by 1C, 1A[16], and 2E, 2G, and 2H[33–35]. SNP type 1 predominates in Thailand[41], Bangladesh, Indonesia, and Philippines[16]. The current phylogeography of *M. leprae* implies that humans brought leprosy to South Asia from other parts of the continent.

**Hypermutated *M. leprae* strains.** Eight *M. leprae* strains (85054, S15, Amami, Zensho-4, Zensho-5, Zensho-9, Br14-3, and Br2016-15), belonging to five different SNP subtypes, had unusually long branches in the MP tree (Fig. 3a) because they contained on average 92 more SNPs than the other strains but approximately the same number of InDels. Comparative analysis revealed one unique feature linking the observed "hypermutated" strains, namely deleterious mutations in the endonuclease III gene *nth* (ML2301) due to frameshifts and premature stop codons (Table 1).

**Drug resistance.** DR-associated SNPs were detected in the DRDR in 24 strains for *folP1*, 11 strains for *rpoB*, and 2 strains for *gyrA* (in bold in Table 1). Previously described mutations were identified in *folP1* at codons 53 ($n = 7$) and 55 ($n = 17$), except in one isolate (Bn8-52), which had mutations at codons 55 and 145. Eleven strains had known mutations that confer rifampicin resistance in their *rpoB*-DRDR, while two strains (Kutatsu-6 and S15) harbor one additional mutation, and one (Br14-3) has two

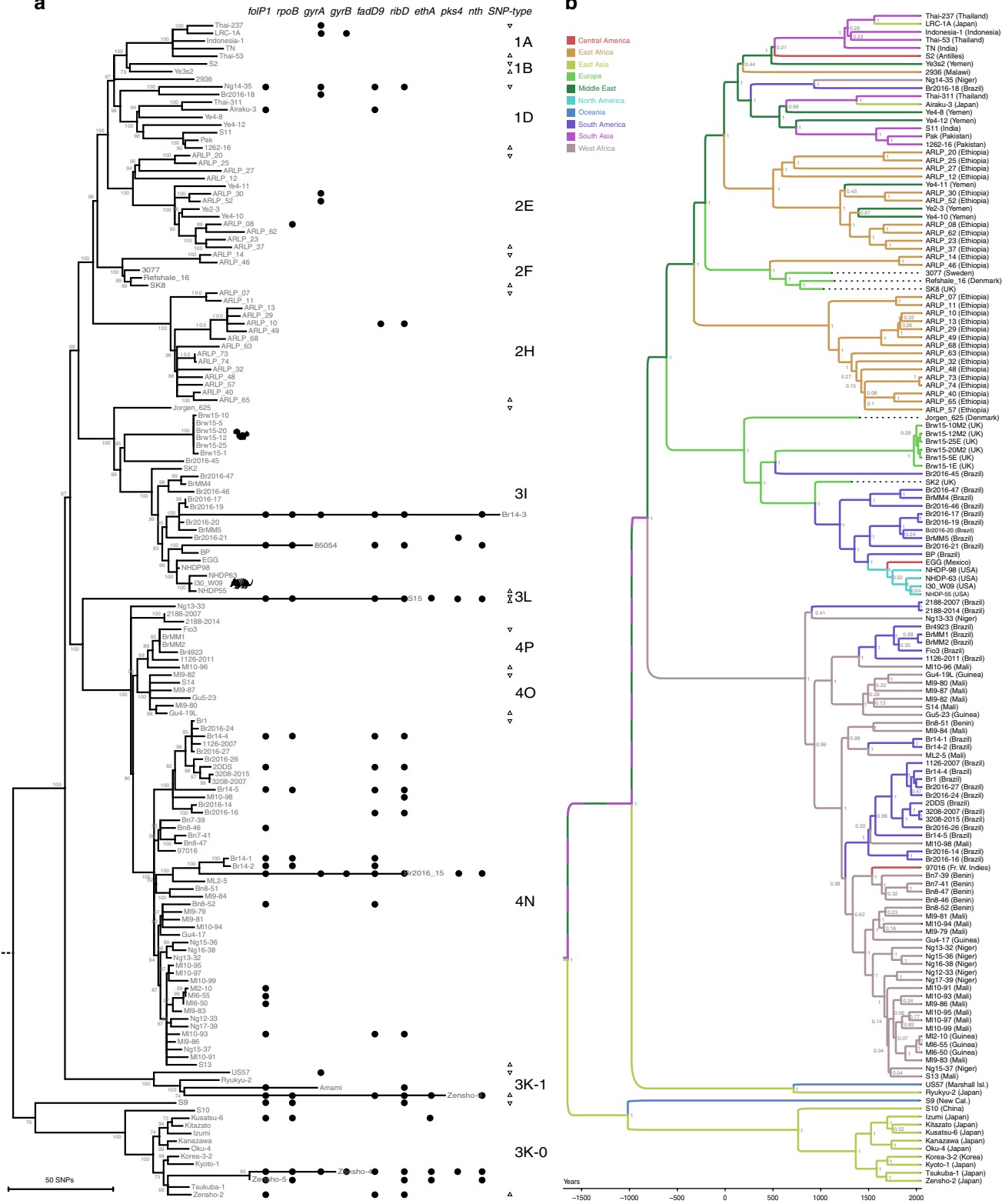

**Fig. 3** Phylogeny of *M. leprae*. **a** Maximum parsimony tree of 154 genomes of *M. leprae*. The tree is drawn to scale, with branch lengths representing number of substitutions. *M. lepromatosis* was used as outgroup. Bootstrap values (500 replicates) are shown next to the branches. Dots indicate protein-changing mutations in the corresponding gene as given in Table 1. **b** Bayesian phylogenetic tree of 146 genomes of *M. leprae* calculated with BEAST 2.4.4. Hypermutated samples with mutations in the *nth* gene were excluded from the analysis. The tree is drawn to scale, with branch lengths representing years of age. Samples were binned according to geographic origin as given in the legend. Posterior probabilities for each node are shown in gray. Location probabilities of nodes were inferred by the Discrete Phylogeny model

**Table 1 Mutations in genes associated with drug resistance**

| Sample | folP1 ML0224 | rpoB ML1891c | gyrA ML0006 | gyrB ML0005 | fadD9 ML0484c | ribD ML1340 | ethA ML0065 | pks4 ML1229 | nth ML2301c |
|---|---|---|---|---|---|---|---|---|---|
| 2188—2007 | . | . | . | . | . | G62D[1] | . | . | . |
| 2188—2014 | . | . | . | . | . | G62D | . | . | . |
| 85054 | **P55L** | **S456L** G52E | . | . | W878* | D256N | . | . | R197* |
| 2DDS | **P55R** | . | . | . | L396P | S58R | . | . | . |
| LRC-1A | . | . | I851T | V214G | . | . | . | . | . |
| Airaku-3 | **T53I** | . | . | . | N304fs | . | . | . | . |
| Amami | **P55L** | . | . | . | . | R236C | . | . | L145fs |
| ARLP_08 | . | H200Y | . | . | . | . | . | . | . |
| ARLP_10 | . | . | . | . | Y927D | G61C | . | . | . |
| ARLP_30 | . | . | G1115R | . | . | . | . | . | . |
| ARLP_52 | . | . | G1115R | . | . | . | . | . | . |
| Bn8-46 | **P55R** | . | . | . | . | . | . | . | . |
| Bn8-52 | **P55L**N145H | . | . | . | W1108fs | . | . | . | . |
| Br14-1 | **P55R** | **S456M** | . | . | G148fs | . | . | . | . |
| Br14-2 | **P55R** | **S456M** | . | . | G148fs | . | . | . | . |
| Br14-3 | **P55L** | **H451Y** G448D T433I | **A91V** | . | Q107* | G94D | . | . | E173* |
| Br14-4 | **P55R** | **S456L** Y171N | . | . | L396P | I56T | . | . | . |
| Br14-5 | **P55R** | **S456L** | . | . | A919E | C222W | . | . | . |
| Br2016-15 | **P55L** | . | V731I | T503I | G796S | A63T | . | M14I | N142fs |
| Br2016-16 | . | . | . | . | L998fs | K267fs | . | . | . |
| Br2016-18 | . | . | S307L | . | . | . | . | . | . |
| Br2016-21 | . | . | . | . | . | . | . | I932fs | . |
| Kusatsu-6 | **P55L** | T433I **D441Y** | . | . | . | . | K477fs | . | . |
| Ml10-93 | **P55R** | . | . | . | R73fs | S58R | . | . | . |
| Ml10-98 | . | . | . | . | . | A10fs | . | . | . |
| Ml2-10 | **T53R** | . | . | . | . | . | . | . | . |
| Ml6-50 | **T53R** | . | . | . | . | . | . | . | . |
| Ml6-55 | **T53R** | . | . | . | . | . | . | . | . |
| Ng14-35 | **P55R** | . | S307L | . | A594T | D34del | . | . | . |
| S15 | **T53I** | G432S **H451**D | . | . | D466N | Q117* | D63N D323N | T334I | G146fs |
| S9 | **T53I** | R791Q | . | . | . | S58N | . | . | . |
| Thai-237 | . | . | I851T | . | . | . | . | . | . |
| US57 | . | . | G362E | . | . | . | . | . | . |
| Zensho-2 | **P55L** | . | . | . | Y562fs | G94D | . | . | . |
| Zensho-4 | **T53I** | P51S **S456L** | **A91V** | D464N | R314C | D77N | A25T | Q1719* | L163fs |
| Zensho-5 | **P55L** | **S456L** P583L | . | . | . | G204C | G390A | . | N142fs |
| Zensho-9 | **P55L** | **H451Y** G681S | . | . | A973T | P150L | P383L | . | E122* |

In bold, substitutions or residues known to confer drug resistance in *M. leprae*; * premature stop codon, fs frameshift, dot (.) wild type. RpoB numbering is based on *M. leprae*, *E. coli* numbering in brackets: 51 (126), 52 (127), 171 (246), 200 (275), 432 (507), 433 (508), 441 (516), 448 (523), 451 (526), 456 (531), 681 (756), 791 (866). [1]Coverage below the threshold, both "2188" isolates come from the same patient but after an interval of 7 years[28]

additional mutations in the DRDR (Table 1). One of these additional mutations (G432S) does not confer rifampicin resistance to recombinant *Mycobacterium smegmatis*[42] whereas no information is available for the remaining three (T433I, G448D, and T508I) except that the G448A substitution does confer rifampicin resistance in *M. tuberculosis*[43]. Also, 5 of the 11 rifampicin-resistant strains had additional missense mutations in *rpoB* (85054, Br14-4, Zensho-4, Zensho-5, and Zensho-9) while 2 strains (ARLP_08 and S9) presented non-synonymous SNPs outside the DRDR (Fig. 4). Compensatory mutations in *rpoA* and *rpoC*, encoding the alpha and beta-prime subunits of RNA polymerase, can occur in rifampicin-resistant *M. tuberculosis*[44]. We found one non-synonymous SNP in *rpoA*, substitution T187P in the rifampicin-resistant strain Br14-5, and seven non-synonymous SNPs in *rpoC* (Supplementary Data 2), of which

two occurred in the drug-resistant strains S15 (A258T) and Zensho-4 (H1133Y).

Two strains had known quinolone resistance mutations in the DRDR of *gyrA* and six harbored different single mutations elsewhere in the gene. Three isolates had a missense mutation in *gyrB*, including two within the DRDR (Table 1). Five strains harbor deleterious mutations in the *ethA* gene, encoding a monooxygenase that activates thioamide prodrugs in *M. tuberculosis*[45,46]. Interestingly, in addition to *ethA* and *nth*, three genes (*fadD9*, *ribD*, *pks4*) were mutated almost exclusively in MDR strains occurring 18, 19, and 4 times, respectively (Table 1).

**Retracing the emergence of drug resistance in leprosy patients.** Prior to the introduction of MDT in the 1980s, patients were treated with dapsone or other antimicrobials as monotherapies of

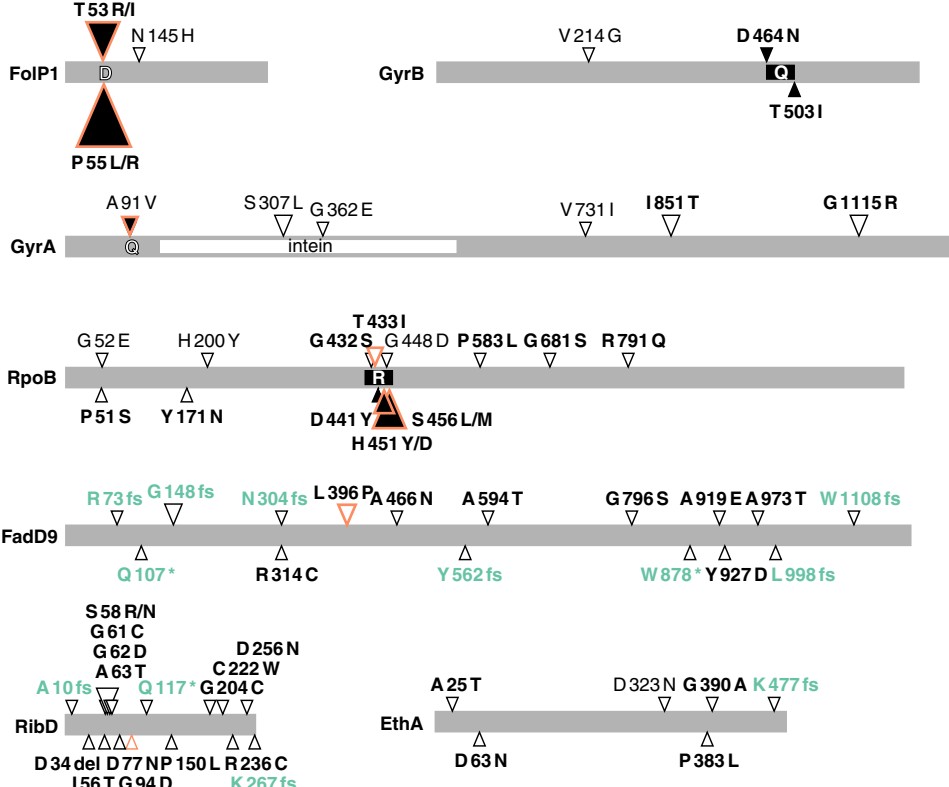

**Fig. 4** Mutations of *M. leprae* genes associated with antimicrobial resistance. Triangles point to the location of the mutation in the protein. Black triangles indicate known resistance-conferring mutations identified in this study that are situated in the drug resistance determining regions (DRDR): D dapsone, Q quinolone, R rifampicin. Orange border means the mutation was found to be homoplasic. Triangle size reflects the number of isolates from this study harboring the mutation, ranging from 1 to 17. Frameshifts and premature stop codons are in turquoise. Substitutions predicted to have an impact on the biological function of the protein[75] are in bold. Proteins are drawn to scale

varying duration[7,8]. Since genomics uncovered new mutations that are associated with antimicrobial resistance in other bacteria, such as those in *ethA* and *gyrB*, this prompted us to try and retrieve the clinical records of six patients whose strains displayed resistance to three or more drugs (dapsone, rifampicin, quinolones, and thioamides). Four of these extensively drug-resistant (XDR) strains were from multibacillary patients in Japan who had received a succession of monotherapies in the pre-MDT era and our genome analysis enabled the chronology of resistance emergence to be retraced. This is illustrated in Fig. 5, and sadly exemplified by the strain from patient Zensho-4 who was diagnosed in 1963 and first treated with protionamide followed by thiambutosin, both of which show cross-resistance and likely require activation by the EthA mono-oxidase that acquired the A25T missense mutation[47]; then treatment began with dapsone leading to emergence of the T53I mutation in *folP1*, followed by rifapentine that selected the S456L mutation in *rpoB*, and continued with ofloxacin, to which resistance arose from the A91V mutation in *gyrA* and D464N in *gyrB*. Molecular drug susceptibility testing was performed in 1998 and the patient finally cured by a regimen comprising clofazimine, minocycline, chloramphenicol and levofloxacin/sparfloxacin. The fifth XDR strain was from a newly diagnosed Brazilian case (Br2016-15) with no history of treatment for leprosy, confirming the ongoing transmission of primary antimicrobial resistance, while details of the sixth case could not be recovered.

**Genes under positive selection.** We also identified genes containing an unusually high number of polymorphisms, multiple alleles, and homoplasies (Supplementary Figure 2), which could

be indicative of positive selection[48]. Strikingly, the distribution of these polymorphic sites around the genome was not random as they were often clustered, especially proximal to either side of the origin of replication (Supplementary Figure 3). Protein-changing mutations were found in 540 genes, with an average of 1.77 mutations per gene (STD 2.12). Table 2 contains a ranking of genes with at least five non-synonymous mutations or regions with one or more homoplasy (excluding VNTRs). The most polymorphic gene by far was *ML0411*[49] encoding the serine-rich antigen, a member of the immunogenic, surface-exposed PPE protein family. Two other known T-cell antigens whose genes display variability are Lsr2 and EsxA (Table 2). Other than *nth*, three other polymorphic genes (*ML1040c*, *ML1750c*, and *ML1512c*) code for proteins that appear to function in nucleic acid or cyclic nucleotide metabolism (Table 2).

**Discussion**

Here we have optimized and applied highly sensitive procedures to extract *M. leprae* DNA directly from human skin biopsies that is suitable for whole-genome sequencing. The resultant genome sequences were analyzed phylogenetically and used to retrace the origin of the leprosy bacillus, and to identify polymorphisms that had been positively selected during evolution. Such polymorphisms might reflect pressure from the human immune system, from MDT or other forces.

It is striking that the ancestral lineages of *M. leprae* predominate in East Asia, although we should keep in mind that Central Asia has been understudied, so it would be interesting to sequence more samples spanning the East–West axis of Asia, including the Middle East, where the 3K genotype is also

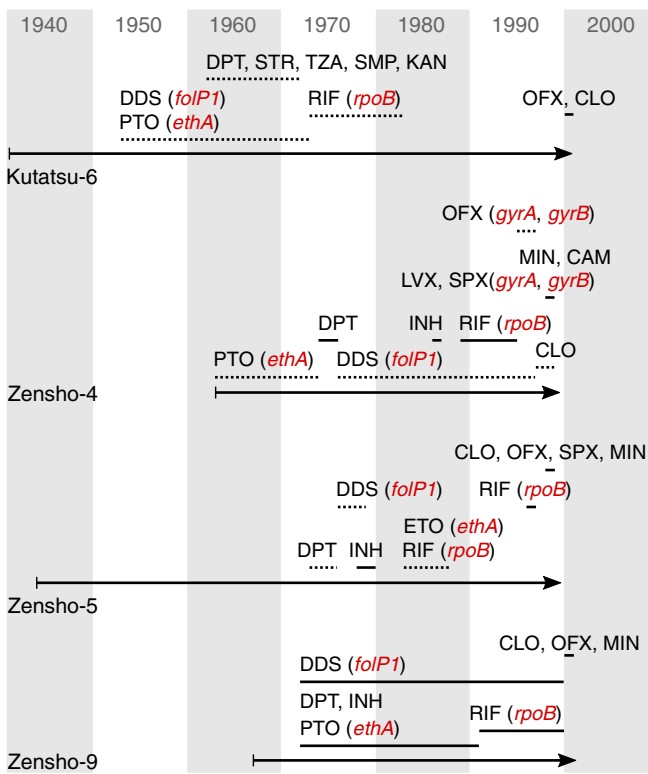

**Fig. 5** Timeline of the leprosy treatment and emergence of drug resistance in the XDR strains. Mutated genes conferring resistance to the corresponding drugs are shown in red. Arrows span from the onset of disease to the end of treatment. Horizontal lines show the period when a drug was given. Dotted lines mean irregular treatment. CAM chloramphenicol, CLO clofazimine, DDS dapsone, DPT thiambutosine (diphenylthiourea), ETO ethionamide, INH isoniazid, KAN kanamycin, LVX levofloxacin, MIN minocycline, OFX ofloxacin, PTO protionamide, RIF rifampicin, SMP sulfamethoxypyridazine, SPX sparfloxacin, STR streptomycin, TZA thiozamin

present[16]. Nevertheless, given the current data on the distribution of the 3K subtype we can deduce that the ancestor of *M. leprae* originated within Eurasia, probably in the Far East.

Endonuclease III (Nth) and the formamidopyrimidine and endonuclease VIII family (Fpg/Nei) of DNA glycosylases are central to the base excision repair pathway in bacteria[50]. Mycobacterial genomes usually contain a single *nth* and two *fpg/nei* genes but *M. leprae* has lost both *fpg/nei* orthologues and retained the *nth* gene. Nth, Fpg, and Nei may have overlapping functions and, in enteric bacteria, mutator phenotypes were observed when *nth* was inactivated in combination with the *fpg* and *nei* genes[51–53]. In *M. smegmatis*, deletion of *nth* and both the *nei* homologs resulted in elevated spontaneous mutation frequencies and increased sensitivity to oxidative stress[54]. Therefore, in the absence of Nei, inactivation of *nth* in *M. leprae* should lead to increased sequence variability, which is consistent with our results.

Strikingly, all *nth* mutants were also drug-resistant so Nth loss likely favors emergence of drug resistance, and *nth* mutations might serve as a surrogate marker for potential drug resistance and treatment failure. A link between a higher mutation rate and drug resistance was observed in strains of *M. tuberculosis* (which has *nth* and two *fpg/nei* genes) belonging to lineage 2, but the molecular basis for this is unknown[55]. For a pathogen with an extremely reduced genome such as *M. leprae*, a hypermutator phenotype could be detrimental and ultimately lethal.

Drug resistance is alarming for leprosy control. There is growing evidence for primary quinolone resistance in strains of *M. leprae* from patients who have never been treated with quinolones for leprosy but may have received this drug for other infections[56]. Five new GyrA mutations were identified in this study, but their effect on FQ resistance remains to be determined. Since two of them arose independently in the GyrA intein, which is removed by protein splicing, they may not impact quinolone activity (Fig. 4). Three non-synonymous mutations were found in *gyrB* (Table 2) and experimental evidence exists for two of them conferring quinolone resistance in in vitro assays or in *M. tuberculosis*[57,58]. To our knowledge, this is the first report of *M. leprae* clinical isolates harboring mutations in *gyrB*. Thus, despite their apparent rarity, mutations in *gyrB* should be systematically assessed in drug resistance screening.

A range of known and new mutations was detected in the DRDR and elsewhere in *rpoB* (Fig. 4; Table 1). Some of these might have a compensatory role in restoring fitness, that is known to be reduced to various degrees in rifampicin-resistant mutants of *M. tuberculosis*[59,60]. Similarly, compensatory mutations in *rpoA* and *rpoC* can occur in rifampicin-resistant *M. tuberculosis*[44]. The *rpoA* substitution T187P in the rifampicin-resistant *M. leprae* strain Br14-5 was shown to be compensatory in *M. tuberculosis*[44]. Rifampicin-resistant isolates of *M. tuberculosis* harbor more mutations in *rpoC* compared to rifampicin-susceptible isolates[44,61]. In our case, we observed no clear correlation between rifampicin resistance and mutations in *rpoC*, which occurred in two resistant and two wild-type strains.

Arguably the most intriguing finding of the present investigation was the remarkably high frequency of mutations in the *fadD9* and *ribD* genes and in 19/23 cases these occur in strains that have at least one mutation that is associated with resistance to a leprosy drug (Fig. 4; Table 1). Functional information for *fadD9* is scarce and the mutations found are predicted to either abolish protein production (8/16) or to cause detrimental amino acid changes (Fig. 4). In the case of *ribD*, 14 different missense mutations were found in a group of 17 variant alleles, indicating that this is likely an essential function. From studies with *M. tuberculosis* it is known that *ribD* encodes an alternative dihydrofolate reductase, with relatively low activity compared to that conferred by the bona fide dihydrofolate reductase gene, *dfrA*[62]. In clinical isolates of *M. tuberculosis*, a promoter mutation causes overexpression of *ribD* that is associated with resistance to the old drug, *para*-amino salicylic acid (PAS), and to certain DHFR inhibitors[63]. This suggests that the mutations detected in the *M. leprae ribD* gene may also confer resistance to PAS and support for this is provided by the fact that vadrine (2-pyridyl-(4)-1,3,4-oxydiazolone-(5)-*p*-aminosalicylate) was used as a drug to treat leprosy before dapsone became widely available[64]. It is thus possible that the *ribD* mutations we report here arose nearly 60 years ago following treatment with vadrine or another PAS derivative. Our discovery of these mutations and those in *fadD9* should encourage further experimentation in order to establish their true role and contribution to antimicrobial resistance, especially to clofazimine.

## Methods

**Sample collection.** Samples were taken from leprosy patients as punch biopsies of skin (preserved in 70% ethanol or formalin-fixed and paraffin-embedded (FFPE)), which is standard diagnostic procedure for leprosy, or from mouse foot-pads. Details about the samples used in this study are given in Supplementary Data 1 and below.

Origin of S15: Strain S15 corresponds to strain 92041[65], which was isolated from a lepromatous leprosy patient originally from Martinique. The origin of S15 was erroneously attributed to New Caledonia in Monot et al.[16], and the error was subsequently propagated in several publications[2,17,66,67].

**Table 2 Highly polymorphic genes and genomic regions of *M. leprae***

| Gene or region | Description | Non-synonymous mutations (multi-allele loci) | Synonymous mutations | Homoplasy |
|---|---|---|---|---|
| *ML0411* | Serine-rich antigen | 32 (4) | 1 | 4 |
| *ML1040c* | Putative ATP-dependent helicase | 19 (1) | 0 | 0 |
| *fadD9* | Probable fatty-acid-CoA ligase | 16 | 1 | 1 |
| *ML1750c* | Putative nucleotide cyclase | 17 | 1 | 0 |
| *ML1512c* | Putative ribonuclease J | 17 | 0 | 0 |
| *ribD* | Bifunctional enzyme riboflavin biosynthesis protein | 17 | 0 | 1 |
| *rpoB* | DNA-directed RNA polymerase (beta chain) | 13 (1) | 1 | 3 |
| *ML1753c* | Probable transcriptional regulatory protein | 9 | 0 | 0 |
| *gyrA* | DNA gyrase (subunit A) | 8 | 8 | 1 |
| *ML1300* | Conserved hypothetical protein | 8 | 0 | 0 |
| *nth* | Endonuclease III | 7 | 0 | 1 |
| *ctpC* | Metal cation-transporting *P*-type ATPase C | 7 | 2 | 0 |
| *rpoC* | DNA-directed RNA polymerase (beta chain) | 7 | 5 | 0 |
| *ML2687c* | Probable conserved transmembrane protein | 7 | 0 | 0 |
| *ML1052c* | Conserved hypothetical protein | 7 | 1 | 0 |
| *ML0009* | Hypothetical protein | 6 | 1 | 0 |
| *ML0283* | Cation-efflux transporter component | 6 | 1 | 0 |
| *ML2700* | Conserved transmembrane protein | 6 | 7 | 0 |
| *mfd* | Transcription-repair coupling factor | 6 | 1 | 0 |
| *ppsC* | Phenolphthiocerol and DIM synthesis | 6 | 1 | 0 |
| *ethA* | Activates the pro-drug ethionamide | 5 | 0 | 0 |
| *trpE* | Biosynthesis of tryptophan (at the first step) | 5 | 1 | 0 |
| *pknB* | Transmembrane serine/threonine-protein kinase | 5 | 3 | 0 |
| *fas* | Fatty acid synthase | 5 | 1 | 0 |
| *esxA* | Early secretory antigenic target | 2 | 0 | 1 |
| *lsr2* | Dominant T-cell antigen and stimulates lymphoproliferation. | 4 | 1 | 3 |
| *folP1* | Dihydropteroate synthase | 5 (2) | 1 | 2 |
| *ML1752c* | Conserved hypothetical protein | 2 | 1 | 1 |
| *ppsA* | Phenolphthiocerol and DIM synthesis | 4 | 3 | 1 |
| *ML0803* | Two-component sensor kinase | 2 (1) | 0 | 0 |
| *ML0237* | Conserved hypothetical protein (pseudogene) | NA | 1 | 1 |
| *ML0010c-ppiA* | Intergenic region | NA | 3 | 1 |

Origin of the case Ng14-3: We mention in the Results that West Africa harbors only SNP-type 4 strains. Strain Ng14-35 (subtype 1D) is an exception, but the patient (a native of Nigeria) developed leprosy during a prolonged stay in Libya.

LRC-1A, an unidentified strain from Japan: A sample obtained from the Leprosy Research Center (LRC) in Tokyo, initially labeled as Airaku-2, did not contain the mutations in *folP1* and *rpoB* previously detected in Airaku-2 by PCR sequencing[7]. Therefore, we renamed this sample to LRC-1A (standing for SNP subtype 1A sample from LRC).

**DNA extraction and library preparation**. DNA was extracted from 101 human skin biopsies with known BI using a customized in-house protocol combining host tissue digestion and the QIAmp microbiome kit for host DNA depletion, strong bacterial cell lysis and silica-based purification. Punch biopsies (6 mm) in 70% ethanol were first rehydrated in Hank's balanced solution prior to mincing with scissors. Cells were detached from the tissue by 30 min incubation at 37 °C with a mixture of 0.5 U of collagenase and dispase, followed by incubation at 56 °C with 10 mg/ml of trypsin until complete digestion. Free cells were then suspended in 1 ml of phosphate-buffered saline (PBS) and DNA was extracted using the QIAmp DNA microbiome extraction kit according to the manufacturer's recommendations. Each run of extraction included a batch of five to nine samples and one blank control (500 μl of Hank's balanced solution). The presence of *M. leprae* was assessed by PCR using RLEP primers[2] prior to library preparation. Libraries were prepared from 50 μl of extracted DNA using the Kapa Hyperprep kit as described previously[2,5]. DNA from FFPE samples was extracted using the truXTRACTM FFPE DNA kit (Covaris) as described previously[28]. Libraries prepared from the extracted DNA were used directly for shotgun sequencing. *M. leprae* DNA extraction quality was assessed from the percentage of *M. leprae* DNA present in the library inferred by alignment to the reference genome sequence, with a minimum threshold set at 1%. This threshold was chosen because it yields an average genome coverage of at least 5× per sample in a multiplexed run of 10 samples on one HiSeq 2500 lane (yielding around 20 million reads per sample and 100 bases per read).

**Library enrichment**. Libraries with low *M. leprae* content underwent enrichment using whole-genome tiling arrays as described previously[17]. Briefly, Illumina libraries were hybridized onto custom Agilent SureSelect Capture Arrays containing ca. one million DNA probes (60 bp) spanning the entire *M. leprae* genome (tiled every 4 bp), followed by elution and PCR amplification.

**Sequencing**. Sequencing was performed on Illumina Hi-Seq 2000, Hi-Seq 2500, or Mi-Seq instruments.

**Sequence processing**. We took precautions in analyzing the data to avoid false-positive SNP calls. All raw reads were adapter- and quality-trimmed with Trimmomatic v0.33[68]. The quality settings were "SLIDINGWINDOW:5:15 MINLEN:40". Paired-end (PE) data were additionally processed with SeqPrep (https://github.com/jstjohn/SeqPrep) to merge overlapping pairs. This increases the accuracy of sequence in the overlapping area, avoids problems in estimating coverage and creates longer reads, which facilitates InDel calling. Duplicate reads were omitted from downstream analyses. This is especially important for libraries with insufficient *M. leprae* DNA fragments, which is not uncommon for low BI samples or samples that are difficult to process, like FFPE samples. In these cases, library enrichment with array-capture, or very deep sequencing often produce a high number of duplicate reads (DNA fragments that were sequenced multiple times, seemingly increasing the overall genome coverage), with each read having dozens or even hundreds of copies. Such reads will amplify possible artefacts and sequence errors, resulting in false SNP calls.

**Sequence analysis**. Preprocessed reads were mapped onto the *M. leprae* TN reference genome (GenBank AL450380.1) with Bowtie2 v2.2.5[69]. We filtered out all reads with mapping quality below 8 and omitted repetitive regions in the reference sequence. We also omitted ribosomal RNA (rRNA) genes because alignments in these regions tend to be error prone. This is because rRNA genes are highly conserved in bacteria, so sequences from other species could map to the *M. leprae* reference sequence. This usually happens when the content of *M. leprae* DNA in a sequencing library is scarce and is even more pronounced when libraries with low *M. leprae* content undergo array-capture. However, because lineage-specific mutations were previously observed in the *M. leprae* rrs gene[30], we manually checked the alignments corresponding to the rRNA genes and added the curated results to Supplementary Data 2.

SNP calling was done using VarScan v2.3.9[70]. To avoid false-positive SNP calls the following cutoffs were applied: minimum overall coverage of five non-duplicated reads, minimum of three non-duplicated reads supporting the SNP, mapping quality score >8, base quality score >15, and a SNP frequency above 80%. InDel calling was done using Platypus v0.8.1[71] followed by manual curation. Completed genome sequences of *M. leprae* Br4923 and *Mycobacterium lepromatosis* (GenBank JRPY00000000.1) were aligned against the *M. leprae* TN reference using LAST[72] using the default parameters for the former and the gamma-centroid option for the latter.

**Mixed samples.** A large number of missing values, especially in lineage-specific loci, points to the presence of more than one strain in a sequencing library. Although not thoroughly tested, in our opinion mixed data sets are mostly due to technical problems or contamination because in some cases we were able to identify the problematic strains. The possible presence of multiple *M. leprae* strains in single skin lesions was not tested in this study, but we expect it to be extremely low. Overall, a few mixed data sets were detected and some were removed from this study, except for samples that we deemed important and describe below. Nevertheless, results were not biased because loci with mixed alleles were treated as missing values.

Zensho-4 seems to contain a fraction of another strain (possibly around 40%) that is closely related to it. Only a few loci had mixed alleles, and these include the A91V substitution in *gyrA* (supported by 62% of reads) and the D464N substitution in *gyrB* (supported by 41% of reads). The latter was attributed to Zensho-4 for simplicity. Similarly, Zensho-5 seems to contain around 30% of Zensho-4. This is the main reason why we could not detect SNPs specific only to Zensho-5 (Fig. 3a), since such SNPs would be "diluted" with wild-type alleles from Zensho-4 and could not pass the SNP "purity" threshold. We included these two samples in this study because they are multi-drug-resistant and belong to the SNP-type 3K-0. Furthermore, mutations in genes conferring drug resistance from this study match with those from earlier reports of these samples, confirming their identity[73].

Thai-311 contains <20% of an unidentified 3K-0 strain that belongs to the Kyoto-1/Zensho-5 cluster of strains (Fig. 3a). SNP calling was not significantly affected. Finally, sample Ye2-3 contained around 25% of an unidentified strain belonging to SNP-type 4. Because we only have few samples from Yemen, we decided to keep Ye2-3 in this study.

**Phylogeny and dating analysis.** Concatenated SNP alignments were used for the analyses. MP trees were constructed in MEGA6[74] using 500 bootstrap replicates. Sites with missing data were partially deleted (80% coverage cutoff), resulting in 3046 variable sites used for the tree calculation. The Subtree-Pruning-Regrafting algorithm was used as the MP search method. Dating analysis and discrete phylogeography were done using BEAST2 v2.4.4[40]. Details are given in the Supplementary Note.

**Data availability.** Sequence data are available from the NCBI Sequence Read Archive (SRA) under accession number SRP072827. Accession numbers for all samples used in this study are given in the Supplementary Data 1. Other relevant data supporting the findings of the study are available in this published article and its Supplementary Information files, or from the corresponding author upon request.

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

## Acknowledgements

We thank the Genomic Technologies Facility at the University of Lausanne for Illumina sequencing and technical support and all the patients and clinical staff who participated in the study. This work was supported by the Fondation Raoul Follereau, the Swiss National Science Foundation grant IZRJZ3_164174, the Swiss Cooperation and Development Center (CODEV), the Heiser Program of the New York Community Trust for Research in Leprosy (grant numbers P15-000827 and P16-000976), and grants CNPq 428964/2016-8 and CAPES PROAMAZONIA 3288/2013.

## Author contributions

S.T.C., P.S., C.A., and A.B. designed the study. C.A., P.S., S.G., A.N.B.F., and P.B. processed the samples, extracted DNA, and prepared sequencing libraries. A.B. and C.L. processed the data. A.B. analyzed the data and prepared figures and tables. A.B., C.A., and S.T.C. interpreted the results and wrote the manuscript with input from other authors. S.G., Y.M., M.N., K.B., C.G.S., M.B.S., R.C.B., M.A.C.F., F.B.F., J.G.B., J.A.C.N., S. B.-S., A.L., A.R.A.-S., Y.A.-Q., A.S.A., G.B., L.V.C., F.S., C.R.J., M.Ko., A.F., S.O.S., M.G., O.K., M.M.A.S., G.O.P., P.N.S., E.N.S., M.O.M., P.S.R., I.M.F.D.B., J.S.S., A.A., M.M., and M.Ka. participated in identification of leprosy cases, patient management, sample collection and preparation, and microscopy.

## Additional information

**Competing interests:** The authors declare no competing financial interests.

Andrej Benjak [1], Charlotte Avanzi[1], Pushpendra Singh[1,2], Chloé Loiseau [1,3], Selfu Girma[4], Philippe Busso[1], Amanda N. Brum Fontes[5], Yuji Miyamoto[6], Masako Namisato[7], Kidist Bobosha[4], Claudio G. Salgado[8], Moisés B. da Silva[8], Raquel C. Bouth[8], Marco A.C. Frade[9], Fred Bernardes Filho[9], Josafá G. Barreto[10], José A.C. Nery[11], Samira Bührer-Sékula[12], Andréanne Lupien[1], Abdul R. Al-Samie[13], Yasin Al-Qubati[13], Abdul S. Alkubati[13], Gisela Bretzel[14], Lucio Vera-Cabrera[15], Fatoumata Sakho[16], Christian R. Johnson[17], Mamoudou Kodio[18], Abdoulaye Fomba[18], Samba O. Sow[18], Moussa Gado[19], Ousmane Konaté[19], Mariane M.A. Stefani[12], Gerson O. Penna[20], Philip N. Suffys[5,21], Euzenir Nunes Sarno[11], Milton O. Moraes[11], Patricia S. Rosa[22], Ida M.F.Dias Baptista[22], John S. Spencer[23], Abraham Aseffa [4], Masanori Matsuoka[6], Masanori Kai[6] & Stewart T. Cole[1]

[1]Global Health Institute, Ecole Polytechnique Fédérale de Lausanne, Lausanne 1015, Switzerland. [2]Department of Microbiology and Biotechnology Centre, Maharaja Sayajirao University of Baroda, Vadodara 390002, India. [3]Swiss Tropical and Public Health Institute, 4051 Basel, Switzerland. [4]Armauer Hansen Research Institute, PO Box 1005, Addis Ababa 1000, Ethiopia. [5]Laboratory of Molecular Biology Applied to Mycobacteria, Oswaldo Cruz Institute, Fiocruz, Rio de Janeiro 21040-360, Brazil. [6]Leprosy Research Center, National Institute of Infectious Diseases, Higashimurayama, Tokyo 189-0002, Japan. [7]AUEN Polyclinic, Nakatomi, Tokorozawa 359-0002 Saitama Prefecture, Japan. [8]Laboratório de Dermato-Imunologia, Instituto de Ciências Biológicas, Universidade Federal do Pará, Marituba 67200-000 Pará, Brazil. [9]Dermatology Division, Ribeirão Preto Medical School, University of São Paulo, Ribeirão Preto 14049-900 São Paulo, Brazil. [10]Spatial Epidemiology Laboratory, Federal University of Pará, Castanhal 68746-360 Pará, Brazil. [11]Leprosy Laboratory, Oswaldo Cruz Institute, Fiocruz, Rio de Janeiro 21040-900, Brazil. [12]Tropical Pathology and Public Health Institute, Federal University of Goiás, Goiânia 74605-050, Brazil. [13]Ministry of Health and Population, c/o National Leprosy Elimination Programme, Taiz, Yemen. [14]Division of Infectious Diseases and Tropical Medicine, University Hospital, Ludwig-Maximilians-University, 80802 Munich, Germany. [15]Laboratorio Interdisciplinario de Investigación Dermatológica, Servicio de Dermatología, Hospital Universitario, Universidad Autónoma de Nuevo León, 64460 Monterrey, Mexico. [16]Programme National Lèpre de Guinée, Conakry, Guinea. [17]Centre Interfacultaire de Formation et de Recherche en Environnement pour le Développement Durable, University of Abomey-Calavi, 03 BP 1463 Jericho, Cotonou, Benin. [18]Centre National d'Appui à la Lutte Contre la Maladie, Bamako, Mali. [19]Programme National de Lutte contre la Lèpre, Ministry of Public Health, Niamey, Niger. [20]Tropical Medicine Centre, University of Brasília, Fiocruz, Brasília 70910-900, Brazil. [21]Department of Biomedical Sciences, Mycobacteriology Unit, Tropical Institute of Medicine, 2000 Antwerp, Belgium. [22]Lauro Souza Lima Institute, Bauru 17034-971 São Paulo, Brazil. [23]Department of Microbiology, Immunology, and Pathology, Colorado State University, Fort Collins, CO 80523-1682, USA

