## [Peer Review File · Nature Communications]

Reviewer #1 (Remarks to the Author):

This is an interesting and well written paper that not just extends our understanding of the phylogenomics of *M. leprae* but brings in surprising new angles on genome evolution in this organism, particularly the discovery of hypermutator strains which are more prone to develop AMR and some intriguing new mutations that appear to be linked to resistance. In addition, it brings genomics into the clinical area for leprosy, with direct sequencing from patient samples. As a result, this paper is likely to be of interest to a wide range of readers from bioinformaticians and genome biologists all the way through to clinicians caring for leprosy patients.

I have one minor query: they say that the most polymorphic gene by far was ML0411, but as this is highly repetitive, at least at the protein level, can they be confident that their assemblies and SNP predictions work well in this context?

Reviewer #2 (Remarks to the Author):

This work represents a significant advance in our understanding of the evolution and the emergence of drug resistance of the unculturable pathogen *M. leprae*, which remains a public health problem in different continents. By customizing a protocol for DNA extraction directly from patients' skin biopsies, thereby reducing the need for cumbersome DNA array capture-based enrichment procedures, the authors could obtain 120 new genome sequences. While broadly in line with previous data on the population structure of the bacterium, phylogenetic analysis of these genomes and 34 from previous studies originating from 25 countries reveals the existence of a second ancestral lineage in East Asia and Micronesia, further supporting the hypothesis that the disease emerged within this world region.

A catalogue of gene mutations known or predicted to confer drug resistance was also identified in classical target genes, as well as newly described mutations in two other genes strongly associated with genotypic resistance profiles, suggesting their potential involvement in the acquisition of drug resistance. Retrieval of old clinical records allowed the authors to propose a plausible scenario of sequential acquisition of drug resistance conferring mutations for patient cases under longitudinal treatment.

More unexpected is the finding of hypermutated strains in distinct lineages, in close association with drug resistance profiles and all linked to deleterious mutations in the endonuclease III gene *nth*.

These new findings are of great interest for the field. However, some weaknesses should be addressed in order to clarify some limitations and implications, and to further exploit the obtained data.

Major points:

1. Although the technical challenge of sequencing an unculturable organism is certainly acknowledged, the quality of the obtained genome sequences needs additional description. The 5 X coverage depth that was accepted as a minimum for SNP analysis is extremely low. According to Supplementary Information, 40 out of the 154 strains have coverage depths below 30X. This would be considered below minimal thresholds for genome wide SNP analysis in other, less challenging bacteria. As a result, numerous regions are probably not minimally covered in a substantial number of genomes. The portions of the reference genome minimally covered in individual strains should be indicated, as well as the total number of genome positions without or with at most 20% missing data used for the SNP analysis and for the maximum parsimony trees. Potential biases in the phylogenetic reconstruction resulting from relatively large genome portions not represented in some strains or strain groups need to be considered and discussed.
2. While the TMRCA of around 3,700 years of all *M. leprae* strains appears robust to variations in the Bayesian models tested, the overall precision in the dating of branches within the Bayesian tree is more questionable. From what is seen for 3I genotypes, time of introductions from Europe into Brazil that can be inferred by the tree correspond to pre-Columbian periods (around 500 and

950 A.D.), which is hard to reconcile with the now generally accepted scenario of the first contacts between Europe (or Africa) and South America. The same question arises e.g. for the node shared between Br2016-18 from Brazil and a sample from Niger, also predating 1000 A.D., and even more so for the node shared by sample S2 from Antilles and samples from South Asia, dating back to around 500 A.D. Highest posterior density intervals around the different nodes should be calculated and represented in order to estimate the precision. A comment should be added in accordance.

3. To my knowledge, this is the first time that *M. leprae* genome sequences are obtained from longitudinal samples collected over different disease episodes from same patients, offering a unique opportunity to obtain first insights into the short-term evolution rate within the host. Information in Supplementary Table 1 suggests that such paired samples were analysed for (at least?) three patients, but this information is not exploited, also with respect to its potential consistency with the long-term mutation rate estimated on the basis of the Bayesian analysis of the global strain population. Were there any mutations identified between samples within a pair? Even if inherent stochasticity and the limited sample size limit conclusions, the consistency of the obtained results with the Bayesian estimate would be worthy a comment.

4. The general discussion could be improved. While plausible hypotheses are offered for explaining the remarkably frequent mutations in *fadD9* and *ribD* genes, the discussion around mutations in genes canonically associated with mycobacterial drug resistance (*rpoB*, *gyrA*, *gyrB*) and compensation of fitness cost (*rpoA* and *rpoC*) appears less exciting. The observation of hypermutated strains in connection with mutations in the *nth* gene is not further discussed. Yet, such hypermutation features, predictably boosting diversification and adaptive potential, come like a surprise in an otherwise so genetically conserved *Mycobacterium* that has undergone massive genome decay. Their independent emergence in different strain lineages in exclusive association with drug resistant cases inevitably raises the question on causality, i.e. were they positively selected as a result of or independently from drug pressure? Also, did these genomes also contain more mutations of other types than SNPs (i.e. indels)? Could hypermutation rates be estimated relatively to the estimated mutation rate of normal strains? And last but not least, do these findings have some implications for the ongoing debate around the potential existence of hypermutable strains of *M. tuberculosis* having a higher propensity for acquiring drug resistance (see e.g. Ford et al., *Nature Genetics*, 2013)? As the latter *Mycobacterium* has *fpg/nei* genes, another mechanism should then probably be invoked, irrespectively of the potential presence of mutations in *nth* in some strains.

Minor points:

1. Results, line 127. For the cases with supposedly drug susceptible strains, please clarify if the disease history was completely unknown or the history was known but not suggestive of defective response to treatment.

2. Results, line 149. In contrast to what the text suggests, the presence of SNP type 2 in India and Nepal is not corroborated by the phylogeny in Fig. 3, as this region is neither represented among SNP type 2 samples, nor among location probabilities of corresponding nodes. India and Nepal should be removed from this statement.

3. Results, lines 162-163. The broad diversity of genotypes from Brazil in general, which extends beyond genotype 3I, would need an additional explanation.

4. Results, line 178. Specify per site per year for the substitution rate.

5. Methods, line 333. Which procedures and kits were used to prepare the DNA libraries?

6. Methods, line 370. Was the omission of repetitive regions based on annotation information or based on self-self BLAST analysis of the reference? Such omission needs to be described more precisely. This would also help understanding how the regions subject to positive selection included ML0411, a member of the PPE protein family.

7. Figure 2. It would be useful to distinguish archeological from contemporary samples on the map.

8. Figure 3. Specify that the values shown next to the branches correspond to posterior probabilities of nodes. It is difficult to see line thickness differences and their correspondence with location probabilities.

9. Figure 4. Orange border is likely missing for one *RibD* mutation.

10. Figure 5. CAM, chloramphenicol is lacking in the legend.

Reviewer #3 (Remarks to the Author):

The authors describe a novel and important practical approach to isolate low amounts of bacterial DNA directly from clinical specimens from patients affected with leprosy. Importantly, the quality of the material allowed derivation of reliable whole genomic sequences of the bacterium. There was a significant correlation between *M. leprae* genome coverage and clinical bacillary loads, determined in material from the same patient's lesions by conventional microscopy. This is a very useful new approach that can find wider application in the field of (bacterial) infectious diseases.

The new technique made it possible to perform comparative phylogenetics using over 150 sequenced *M. leprae* genomes obtained from 25 countries. This allowed identification of new *M. leprae* lineages and tracing of their evolution. Furthermore, attempts were made to correlate novel mutations with drug resistance phenotypes.

While the first part is important and convincing, the part on antibiotic resistance is somewhat less clear and more speculative in a number of cases. A series of novel mutations is identified in *M. leprae* genes, including genes previously found to be responsible for drug resistance. Whereas in the latter cases their correlations with drug resistance seem plausible, such as in the convincing case of patient Zensho-4 (page 11), other -mostly newly identified- mutations may or may not be causal to the drug resistance phenotypes. For example, the authors state on page 10 line 214 etc. that no information is available for most of the new mutations found (T433I, G448D and T508I). Another example is line 217 etc: although the additional missense mutations in *rpoB* (85054, Br14-4, Zensho-4, Zensho-5 and Zensho-9) are certainly of interest and likely to be causal to rifampin resistance confirmatory evidence is lacking in these cases, either experimentally or from other pathogens. For patient Br2016-15 (page 11, line 249) limited evidence is given, for the sixth case no records could be traced. This limited confirmatory evidence makes parts of the section Retracing the emergence of drug resistance in leprosy patients somewhat speculative, because it seems that for only some mutations a firm correlation with drug resistance has been established. One can therefore only agree with the last sentence of the discussion.

Perhaps one of the most interesting findings is the *nth* excision repair mutations, because *M. leprae* contains only one *nth* copy, having deleted *fpg* and *nei*. The deleterious mutations in *nth* were predicted to lead to hypermutated *M. leprae*, which was confirmed in the 8 strains investigated: all 8 strains were drug resistant. This is a plausible explanation, because hypermutations as a result of impaired *nth* activity are likely to have caused the phenotype. Moreover, rather convincing experimental evidence to support this exists from *M. smegmatis* and other bacteria. I am not sure why the authors did not cause to highlight this important finding somewhat more.

Minor comments

Page 10 line 207: it would be helpful to indicate whether these 24 strains were DR

Fig 4 legend: does not explain the difference between open and closed triangles.

Page 11, line 236-237: I was unable to find in the suppl. note (referred to on page 14 line 322) any information reg. their drug resistance.

Reviewers' comments are in black.

Authors' answers are in blue.

Reviewer #1

This is an interesting and well written paper that not just extends our understanding of the phylogenomics of *M. leprae* but brings in surprising new angles on genome evolution in this organism, particularly the discovery of hypermutator strains which are more prone to develop AMR and some intriguing new mutations that appear to be linked to resistance. In addition, it brings genomics into the clinical area for leprosy, with direct sequencing from patient samples. As a result, this paper is likely to be of interest to a wide range of readers from bioinformaticians and genome biologists all the way through to clinicians caring for leprosy patients.

I have one minor query: they say that the most polymorphic gene by far was *ML0411*, but as this is highly repetitive, at least at the protein level, can they be confident that their assemblies and SNP predictions work well in this context?

The nucleotide sequence of *ML0411* is not repetitive (we checked it using the Tandem Repeat Finder Program (Nucleic Acid Research 1999, G. Benson)). Moreover, we manually checked the alignments in this gene (and in any other region with high SNP abundance or regions conferring DR for that matter) to ensure that there were no erroneous SNP calls derived from possible alignment problems. Thus, we are confident that the reported SNPs and indels for *ML0411* are accurate.

Reviewer #2

This work represents a significant advance in our understanding of the evolution and the emergence of drug resistance of the unculturable pathogen *M. leprae*, which remains a public health problem in different continents. By customizing a protocol for DNA extraction directly from patients' skin biopsies, thereby reducing the need for cumbersome DNA array capture-based enrichment procedures, the authors could obtain 120 new genome sequences. While broadly in line with previous data on the population structure of the bacterium, phylogenetic analysis of these genomes and 34 from previous studies originating from 25 countries reveals the existence of a second ancestral lineage in East Asia and Micronesia, further supporting the hypothesis that the disease emerged within this world region.

A catalogue of gene mutations known or predicted to confer drug resistance was also identified in classical target genes, as well as newly described mutations in two other genes strongly associated with genotypic resistance profiles, suggesting their potential involvement in the acquisition of drug resistance. Retrieval of old clinical records allowed the authors to propose a plausible scenario of sequential acquisition of drug resistance conferring mutations for patient cases under longitudinal treatment.

More unexpected is the finding of hypermutated strains in distinct lineages, in close association with drug resistance profiles and all linked to deleterious mutations in the endonuclease III gene *nth*.

These new findings are of great interest for the field. However, some weaknesses should be addressed in order to clarify some limitations and implications, and to further exploit the obtained data.

Major points:

1. Although the technical challenge of sequencing an unculturable organism is certainly acknowledged, the quality of the obtained genome sequences needs additional description. The 5 X coverage depth that was accepted as a minimum for SNP analysis is extremely low. According to Supplementary Information, 40 out of the 154 strains have coverage depths below 30X. This would be considered below minimal thresholds for genome wide SNP analysis in other, less challenging bacteria. As a result, numerous regions are probably not minimally covered in a substantial number of genomes. The portions of the reference genome minimally covered in individual strains should be indicated, as well as the total number of genome positions without or with at most 20% missing data used for the SNP analysis and for the maximum parsimony trees. Potential biases in the phylogenetic reconstruction resulting from relatively large genome portions not represented in some strains or strain groups need to be considered and discussed.

The reviewer raised important points that must be addressed in more detail. We are aware that the SNP calling thresholds were adjusted to accommodate a number of modestly covered samples. However, we took some precautions to avoid false SNP calls (the threshold for SNP calling was the following: duplicated reads are omitted, total coverage must be at least 5, variant frequency at least 80%, min supporting reads at least 3, base quality must be above 15, and p value

below 0.05). Nevertheless, the reviewer is right in pointing out that the fraction of the genome covered above 5x and the number of missing SNP calls are more important metrics than the average genome coverage alone. We added this information in Table S1. Only six samples covered less than 90% of the reference genome with at least 5 non-duplicate reads, the worst (sample 1126-2007) covering 67% of the genome. Only five samples had over 10% missing calls in the SNP table, the worst being 1126-2007 with 28% missing calls. We do not think that this caused a wrong placement of those samples in the Maximum Parsimony tree for two main reasons: 1) their phylogeny was still estimated based on hundreds of SNP that were properly called, and 2) the topology from the Bayesian analysis is virtually identical to that of the Maximum Parsimony (BEAST takes into account sites with missing values and estimates their most likely state, which is used for tree reconstruction). Finally, the same arguments can be used to answer the question about the removal of (possibly too many) informative sites in the Maximum Parsimony analysis because of the 20% missing value threshold. Only six sites (out 3056) were omitted due to this threshold.

2. While the TMRCA of around 3,700 years of all *M. leprae* strains appears robust to variations in the Bayesian models tested, the overall precision in the dating of branches within the Bayesian tree is more questionable. From what is seen for 3I genotypes, time of introductions from Europe into Brazil that can be inferred by the tree correspond to pre-Columbian periods (around 500 and 950 A.D.), which is hard to reconcile with the now generally accepted scenario of the first contacts between Europe (or Africa) and South America. The same question arises e.g. for the node shared between Br2016-18 from Brazil and a sample from Niger, also predating 1000 A.D., and even more so for the node shared by sample S2 from Antilles and samples from South Asia, dating back to around 500 A.D. Highest posterior density intervals around the different nodes should be calculated and represented in order to estimate the precision. A comment should be added in accordance.

We fear that the reviewer is misinterpreting the dating tree. Node dates do not indicate times of introductions. For example, if a strain were introduced into a new place now, that strain would still carry the same ancestral information and its TMRCA would be much older than the time of its introduction into the new place. With the argument about the Brazilian 3I strains, the reviewer came close to a possible scenario where the TMRCA should agree with the time of introduction, which is the case of a single introduction. For example, if only one 3I strain had ever been introduced into the Americas, we would expect the TMRCA of all Brazilian 3I strains to be younger than the colonization of America. It is hard to believe that despite the massive immigrations that occurred from different parts of the world and lasted for centuries, only one strain, from every major lineage, was introduced into America. Rather, a large diversity of *M. leprae* was simply “translocated” there by numerous introductions from different areas.

We agree with the reviewer about highest posterior density (HPD) intervals. Because these are difficult to fit in the main figure, we created a new supplementary figure (S1) with the same tree showing additional metrics.

3. To my knowledge, this is the first time that *M. leprae* genome sequences are obtained from longitudinal samples collected over different disease episodes from same patients, offering a unique opportunity to obtain first insights into the short-term evolution rate within the host. Information in Supplementary Table 1 suggests that such paired samples were analysed for (at least?) three patients, but this information is not exploited, also with respect to its potential consistency with the long-term mutation rate estimated on the basis of the Bayesian analysis of the global strain population. Were there any mutations identified between samples within a pair? Even if inherent stochasticity and the limited sample size limit conclusions, the consistency of the obtained results with the Bayesian estimate would be worthy a comment.

Details about those three patient samples were published in Stefani *et al.* 2017, PLOSntd (but with no dating analysis), which we added in the current manuscript. Serial samples 3208–2007 and 3208–2015 differed in only two SNPs, and the estimated mean TMRCA was 25 ya (95% HPD 0.86-60 ya), while samples 2188–2007 and 2188–2014 were identical and their TMRCA was 20 ya (95% HPD 0.86-65 ya). Note that the 95% HPD range will always overlap with the real sample dates because those were given as priors. The 95% HPD range was relatively narrow, given the very weak genetic signal for such a short timeframe and the fact that the dating is largely based on centuries-old samples. As the reviewer pointed out, we did not comment on dating results of those samples because of the limited sample size, the short time span, and the fact that the mutation rate under antibiotic pressure is likely biased (selective pressure, bottlenecks), and therefore not representative for the entire population, so there is little we could say conclusively.

4. The general discussion could be improved. While plausible hypotheses are offered for explaining the remarkably frequent mutations in *fadD9* and *ribD* genes, the discussion around mutations in genes canonically associated with mycobacterial drug resistance (*rpoB*, *gyrA*, *gyrB*) and compensation of fitness cost (*rpoA* and *rpoC*) appears less exciting. The observation of hypermutated strains in connection with mutations in the *nth* gene is not further discussed. Yet, such hypermutation features, predictably boosting diversification and adaptative potential, come like a surprise in an otherwise so genetically conserved *Mycobacterium* that has undergone massive genome decay. Their independent emergence in different strain lineages in exclusive association with drug resistant cases inevitably raises the question on causality, i.e. were they positively selected as a result of or independently from drug pressure?

Since all the *nth* mutants were also DR, we argue that disruption of *nth* probably causes a significant fitness cost that would be prohibitive under neutral conditions. However, the idea that *nth* mutations might directly confer a level of resistance to some of the antileprosy drugs cannot be excluded, although at the

moment, we do not have a plausible hypothesis. We think that the *nth* case of *M. leprae* is a good example of the plasticity vs. stability evolutionary dilemma in genome survival, similar to what has been observed with transposon activity in other species for example.

Also, did these genomes also contain more mutations of other types than SNPs (i.e. indels)?

This is a good question. Indel content is identical to the other strains, which is in agreement with the function of *nth*. We added the information about the indel content in the results on p.9 "...because they contained on average 92 more SNPs than the other strains but approximately the same number of InDels."

Could hypermutation rates be estimated relatively to the estimated mutation rate of normal strains?

In theory yes, but there are a couple of issues. We do not know the exact time these strains were evolving as *nth* mutants. Some of the patients that were infected with an *nth* mutant have a long history of recurrence, spanning a couple of decades, whereas for others we don't know. One of the *nth* strains is probably a primary infection (deduced from the patient's history), which means the mutant must have arisen in another patient. The best way to approximate the mutation rate would be to take into account the number of extra mutations in the *nth* mutant (beyond the average genetic distance, or perhaps the distance from the closest strain) and divide it by the number of years we suspect the *nth* mutant was evolving. But since the latter is uncertain, this would result in a very crude estimate, which we did not feel to be worth pursuing in this paper. Nevertheless, this is an interesting topic for another study with a bigger sample size.

And last but not least, do these findings have some implications for the ongoing debate around the potential existence of hypermutable strains of *M. tuberculosis* having a higher propensity for acquiring drug resistance (see e.g. Ford et al., Nature Genetics, 2013)? As the latter Mycobacterium has *fpg/nei* genes, another mechanism should then probably be invoked, irrespectively of the potential presence of mutations in *nth* in some strains.

As the reviewer indicates, it is difficult to compare *M. leprae* and *M. tuberculosis* due to the much larger gene set in the latter and the contribution of the *fpg/nei* genes. We do now briefly discuss the findings of Ford *et al.* as follows:

"A link between a higher mutation rate and drug resistance was observed in strains of *M. tuberculosis* (which has *nth* and two *fpg/nei* genes) belonging to lineage 2, but the molecular basis for this is unknown⁵⁵."

In addition, we have expanded the discussion on *nth*, at the suggestion of reviewer 3 (see below) and, in the interest of coherence, have moved the corresponding narrative from the results section to the discussion of the revised manuscript.

Minor points:

1. Results, line 127. For the cases with supposedly drug susceptible strains, please clarify if the disease history was completely unknown or the history was known but not suggestive of defective response to treatment.

Information was added to page 6, l. 127: "(87 were from confirmed primary cases, while disease history was unknown for the others).

2. Results, line 149. In contrast to what the text suggests, the presence of SNP type 2 in India and Nepal is not corroborated by the phylogeny in Fig. 3, as this region is neither represented among SNP type 2 samples, nor among location probabilities of corresponding nodes. India and Nepal should be removed from this statement.

We removed India and Nepal from the sentence, and added the references about the presence of SNP subtype 2 in South Asia in the following sentence.

3. Results, lines 162-163. The broad diversity of genotypes from Brazil in general, which extends beyond genotype 3I, would need an additional explanation.

We added a sentence and a citation to this paragraph to explain the broad diversity in Brazil- "Brazil, as expected, contains several *M. leprae* lineages, with the SNP type 4 and SNP subtype 3I being the most prevalent³⁶."

4. Results, line 178. Specify per site per year for the substitution rate.

We corrected it.

5. Methods, line 333. Which procedures and kits were used to prepare the DNA libraries?

We added additional information to the **DNA extraction and library preparation** section.

6. Methods, line 370. Was the omission of repetitive regions based on annotation information or based on self-self BLAST analysis of the reference? Such omission needs to be described more precisely. This would also help understanding how the regions subject to positive selection included ML0411, a member of the PPE protein family.

The omission was done based on annotated repetitive regions, which also include tandem repeats. This filtering is actually redundant with another threshold that we considered, a mapping quality of 8 and above. The lower the quality, the higher is the ambiguity of the alignment (due to repeats or near-repeats). ML0411 is repetitive and of low complexity at the protein level, but it is not repetitive on the nucleotide level (we also checked it using the Tandem Repeat Finder Program (Nucleic Acid Research 1999, G. Benson)). Moreover, we manually checked the alignments in this gene (and in any other region with high

SNP abundance or regions conferring DR for that matter) to assure there were no erroneous SNP calls derived from possible alignment problems. Thus, we are confident that the reported SNPs and indels for *ML0411* are accurate.

7. Figure 2. It would be useful to distinguish archeological from contemporary samples on the map.

We changed the figure. Ancient *M. leprae* strains are now indicated in blue.

8. Figure 3. Specify that the values shown next to the branches correspond to posterior probabilities of nodes. It is difficult to see line thickness differences and their correspondence with location probabilities.

The figure legend was corrected. We agree with the reviewer that it is difficult to distinguish line thickness differences. This is now better shown in the supplementary figure, as mentioned above.

9. Figure 4. Orange border is likely missing for one RibD mutation.

The figure was corrected.

10. Figure 5. CAM, chloramphenicol is lacking in the legend.

Corrected

Reviewer #3:

The authors describe a novel and important practical approach to isolate low amounts of bacterial DNA directly from clinical specimens from patients affected with leprosy. Importantly, the quality of the material allowed derivation of reliable whole genomic sequences of the bacterium. There was a significant correlation between *M. leprae* genome coverage and clinical bacillary loads, determined in material from the same patient's lesions by conventional microscopy. This is a very useful new approach that can find wider application in the field of (bacterial) infectious diseases.

The new technique made it possible to perform comparative phylogenetics using over 150 sequenced *M. leprae* genomes obtained from 25 countries. This allowed identification of new *M. leprae* lineages and tracing of their evolution. Furthermore, attempts were made to correlate novel mutations with drug resistance phenotypes.

While the first part is important and convincing, the part on antibiotic resistance is somewhat less clear and more speculative in a number of cases. A series of novel mutations is identified in *M. leprae* genes, including genes previously found to be responsible for drug resistance. Whereas in the latter cases their correlations with drug resistance seem plausible, such as in the convincing case of patient Zensho-4 (page 11), other -mostly newly identified- mutations may or may not be causal to the drug resistance phenotypes. For example, the authors state on page 10 line 214 etc. that no information is available for most of the new mutations found (T433I, G448D and T508I). Another example is line 217 etc: although the additional missense mutations in *rpoB* (85054, Br14-4, Zensho-4, Zensho-5 and Zensho-9) are certainly of interest and likely to be causal to rifampin resistance confirmatory evidence is lacking in these cases, either experimentally or from other pathogens. For patient Br2016-15 (page 11, line 249) limited evidence is given, for the sixth case no records could be traced. This limited confirmatory evidence makes parts of the section Retracing the emergence of drug resistance in leprosy patients somewhat speculative, because it seems that for only some mutations a firm correlation with drug resistance has been established. One can therefore only agree with the last sentence of the discussion.

Perhaps one of the most interesting findings is the *nth* excision repair mutations, because *M. leprae* contains only one *nth* copy, having deleted *fpg* and *nei*. The deleterious mutations in *nth* were predicted to lead to hypermutated *M. leprae*, which was confirmed in the 8 strains investigated: all 8 strains were drug resistant. This is a plausible explanation, because hypermutations as a result of impaired *nth* activity are likely to have caused the phenotype. Moreover, rather convincing experimental evidence to support this exists from *M. smegmatis* and other bacteria. I am not sure why the authors did not cause to highlight this important finding somewhat more.

We agree with the reviewer. Our initial intent was to keep the manuscript as concise as possible. We moved the *nth* section to the Discussion, and expanded it,

to give this finding more importance.

Minor comments

Page 10 line 207: it would be helpful to indicate whether these 24 strains were DR

These 24 strains are considered DR solely on the presence of known (experimentally confirmed) DR mutations in *rpoB*, *folP1* or *gyrA*, which is the current standard for leprosy. What we pointed out in the corresponding paragraph is, that apart from the known DR mutations, other novel mutations were observed in *rpoB*, *folP1* or *gyrA*, or in related genes, that might be linked to DR and these should be experimentally assessed in future. For 8 of the 24 strains, phenotypic drug resistance had been confirmed (Matsuoka, 2010, Jpn J Lepr) (Honoré 1993 Int J Lepr).

Fig 4 legend: does not explain the difference between open and closed triangles.

Legend was modified.

Page 11, line 236-237: I was unable to find in the suppl. note (referred to on page 14 line 322) any information reg. their drug resistance.

We added a new column "Drug resistant (based on known DR sites)" in Table S1.

Reviewer #2 (Remarks to the Author):

My concerns and comments were appropriately addressed. The interpretation of nodes in the dating tree was indeed mistaken.